# Instance-based Learning for Knowledge Base Completion

**Wanyun Cui**[1], **Xingran Chen**[2]
Shanghai University of Finance and Economics[1]
University of Michigan[2]
cui.wanyun@sufe.edu.cn, chenxran@umich.edu

## Abstract

In this paper, we propose a new method for knowledge base completion (KBC): instance-based learning (IBL). For example, to answer (Jill Biden, lived city,? ), instead of going directly to Washington D.C., our goal is to find Joe Biden, who has the same lived city as Jill Biden. Through prototype entities, IBL provides interpretability. We develop theories for modeling prototypes and combining IBL with translational models. Experiments on various tasks confirmed the IBL model's effectiveness and interpretability.

In addition, IBL shed light on the mechanism of rule-based KBC models. Previous research has generally agreed that rule-based models provide rules with semantically compatible premises and hypotheses. We challenge this view. We begin by demonstrating that some logical rules represent *instance-based equivalence* (i.e. prototypes) rather than semantic compatibility. These are denoted as *IBL rules*. Surprisingly, despite occupying only a small portion of the rule space, IBL rules outperform non-IBL rules in all four benchmarks. We use a variety of experiments to demonstrate that rule-based models work because they have the ability to represent instance-based equivalence via IBL rules. The findings provide new insights of how rule-based models work and how to interpret their rules.

## 1 Introduction

In knowledge base completion (KBC), the learner attempts to infer new facts about the world from given training facts (head, relation, tail). This problem has intrigued many researchers' interest as it's both a fundamental task of relational data representation learning [16, 33] and a benefit for downstream KB-based applications [34, 39].

One typical way of KBC is to answer (head, relation, ?) or (?, relation, tail). Knowledge graph embedding [4, 10] and rule-based reasoning [17, 38] are two typical KBC methods. Knowledge graph embedding learns low-dimensional vectors for entities and relations. One disadvantage of knowledge graph embedding is its lack of interpretability. Rule-based reasoning provides interpretability by transforming queries into human-readable logical rules. However, the predictions are constrained by the rule length $k$. As $k$ increases, the rule-based search space expands exponentially, increasing the difficulty of mining high-quality rules.

In this paper, we explore an alternative method: instance-based learning (IBL) [15, 26]. Instead of performing explicit generalization, IBL generates predictions by comparing query instances with instances seen in training. Despite its rich history in machine learning, IBL has not seen applied in KBC. In terms of KBC, instead of directly finding the target entity to the query, we ask *which alternative entities share the same value with the query entity and relation?* For example, for the query (Jill Biden, lived city, ?), instead of directly finding *Washington D.C.* from the vector space or by rules, we first identify the entity *Joe Biden*, who has the same lived city as Jill Biden. In this

36th Conference on Neural Information Processing Systems (NeurIPS 2022).

paper, Joe Biden is referred to the **prototype** for that query. To answer the query, we will use the prototype's known fact (i.e., (Joe Biden, lived city, Washington D.C.)).

The main motivation behind IBL is the *low-rank assumption*, which holds that real-world knowledge is interdependent. Such assumption has been popularized with the Netflix challenge [9] and now frequently employed in knowledge graph embedding [10, 30, 37]. In the instance-based setting, we found that the low-rank assumption is significant and clear. For example, Joe Biden and Jill Biden share the same lived city because they have the same marriage event and the same children.

IBL has several advantages over traditional KBC methods: first, it enables interpretability through prototypes. For example, inferring Jill Biden's lived city from Joe Biden rather than Barack Obama makes more sense. Second, in contrast to rule-based reasoning, IBL is not constrained by the $k$-step neighborhood. In addition, the search space is always $O(n\_entity)$. Third, it helps KB curation by ensuring high-quality reasoning as the KB grows dynamically [35]. For example, we can still use Joe Biden to infer Jill Biden's lived city if she moves to another city in the future.

Due to the complexity of the knowledge graph, it is still challenging to find correct prototypes among the massive number of entities. Fortunately, although IBL is a new KBC method, we theoretically show that prototypes have closed-form expressions with the well-studied translational models [4, 12, 27] in Sec 3. The derived IBL model has competitive performance with cutting-edge KBC methods. In addition, we establish a joint optimization theory of the IBL model and the translational model in Sec 4, revealing the concordance in terms of their optimization objectives. The combined model inspired by this theory outperforms existing methods in two out of four benchmarks.

Although IBL appears to be irrelevant to rule-based KBC models, we found IBL shed light on its mechanism (Sec 5). Previous literature has commonly agreed that logical rules provide semantically compatible premises and hypotheses. We show that, some rules need to be interpreted through instance-based equivalence, rather than the semantic compatibility. We call them *IBL rules*. Surprisingly, although IBL rules only occupy a small fraction of the entire rules space, they play a significant role in rule-based reasoning. In Sec 7, we show that IBL rules are more critical than all other rules in all four benchmarks, and can even replace all other rules completely with no effect degradation in two of four benchmarks. This phenomenon challenges the previously common understanding of how logical rules work and provide interpretability. Rule-based models rely much on their ability to represent instance-based equivalence to infer new knowledge. The interpretability of logical rules should be revisited from the perspective of instance-based equivalence. And the IBL method proposed in this paper is a more effective alternative to rule-based methods when modeling the equivalency.

Our contributions are outlined as follows: (1) we explore instance-based learning for KBC. IBL provides interpretability while not being constrained by local neighborhoods. (2) Using translational models, we prove that prototypes have closed-form expressions. (3) We developed a joint optimization theory for IBL and translational models as a foundation for combing different KBC models. Experimental results show that the combined model outperforms existing methods on two out of four benchmarks. [1] (4) We found that rule-based reasoning relies heavily on IBL rules to represent the instance-based equivalence. The results suggest that the mechanism and interpretability of rule-based reasoning should be reassessed in terms of instance-based equivalence.

## 2   Related Work

For a comprehensive survey of knowledge graph completion, we refer readers to [33].

**Learning representations for knowledge via prototypes** Another theory that provides support to our work is the prototype theory. The term prototypes, as initially defined in psychologist Eleanor Rosch's study "Natural Categories" [20], has been widely studied in psychology [19] and cognitive linguistics [28]. Bobrow and Winograd [3] used the prototype theory to the general knowledge representation. The general consensus is that knowledge reasoning is dominated by a recognition process in which new objects and events are compared to stored sets of expected prototypes. Although the prototype theory is widely studied in a variety of domains, our study is the first to apply the prototype theory for KBC to our knowledge. We concretize the theory as the equivalence of query entities to prototypes on specific relations, thus allowing knowledge reasoning.

---

[1] We release code at `https://github.com/chenxran/InstanceBasedLearning`

**Instance-based learning** There is a large body of literature on IBL. The k-nearest neighbors algorithm (kNN), which uses the k-nearest examples as prototypes, is the most representative work. Besides, [18] proposed to use model-based methods to calibrate the prototypes in the instance-based method. In recent years, IBL has also shown its great potentials to few-shot learning [1, 25] and self-supervised learning [32]. One trend of IBL is to leverage metric learning for prototype selection [8, 36], in particular, using neural networks for instance embedding [13, 22]. In this paper, similar to [13, 22], we use trainable entity embeddings to represent prototypes.

## 3 Models

In this section, we first formalize the KBC problem (Sec 3.1). Then we demonstrate how translational models allow prototype modeling with closed-form expressions (Sec 3.2 3.3). We also investigate the relation-awareness property for specific translational representations (Sec 3.4).

### 3.1 Problem: Link Prediction for Knowledge Base Completion

KBs are collections of triplet facts that represent world knowledge (head, relation, tail). We denote the fact in the KB as $\mathcal{D} = \{(h_1, r_1, t_1) \cdots (h_n, r_n, t_n)\}$. As it is impossible to collect all facts, a fundamental problem is to predict the missing facts. In this paper, we formalize the completion task as link prediction. The query format in the test set is $(h, r, ?)$ or $(?, r, t)$.

### 3.2 Translational Models

Translational models are one of the mainstream methods of KBC. The main idea behind translational models is to model relations between entities as a translation from head to tail over the vector space. For a new fact, its plausibility can be calculated by the distance between the translated head entity and the tail entity. We formulate the plausibility of $(h, r, t)$ as below:

$$\mathcal{T}(\mathrm{h}, \mathrm{r}, \mathrm{t}) = \|\mathrm{trans_r(emb(h))} - \mathrm{emb(t)}\| \tag{1}$$

where $emb(\cdot)$ denotes the entity embedding, and $trans_r(\cdot)$ denotes the translation w.r.t. $r$.

There are numerous variants under the translational framework. Here we present some typical models. The feasibility of exploiting these models for IBL are investigated in Sec 3.4.

**TransE [4]** stores the embeddings $emb(\cdot)$ in an embedding matrix. It uses the simplest vector addition as the translation $trans_r(\cdot)$. Its scoring function is:

$$\mathrm{TransE}(\mathrm{h}, \mathrm{r}, \mathrm{t}) = \|\mathbf{e}_\mathrm{h} + \mathrm{r} - \mathbf{e}_\mathrm{t}\| \tag{2}$$

**TransR [12]** extends TransE by adding a relationship matrix $\mathbf{W}_\mathrm{r}$, which maps embeddings of entities from the entity space to the relational space:

$$\mathrm{TransR}(\mathrm{h}, \mathrm{r}, \mathrm{t}) = \|\mathbf{W}_\mathrm{r}\mathbf{e}_\mathrm{h} + \mathrm{r} - \mathbf{W}_\mathrm{r}\mathbf{e}_\mathrm{t}\| \tag{3}$$

**RotatE [27]** is a cutting-edge translational model. It represents the translation via rotations in the complex space (denoted as $\circ$). It has the following scoring function:

$$\mathrm{RotatE}(\mathrm{h}, \mathrm{r}, \mathrm{t}) = \|\mathbf{e}_\mathrm{h} \circ \mathrm{r} - \mathbf{e}_\mathrm{t}\| \tag{4}$$

### 3.3 Modeling Prototypes in IBL

In IBL, given the query entity and relation, our primary goal is to locate the prototype whose value is the same as the query and is present in the training data. We say $p$ is the prototype of $(h, r, ?)$ if $(p, r, ?) = (h, r, ?)$. For example, Joe Biden is a prototype of the query (Jill Biden, lived city, ?), since (Jill Biden, lived city, ?) = (Joe Biden, lived city, ?) = Washington D.C. The prototype of $(?, r, t)$ is defined similarly.

We show that the prototype computation is *tractable* using translational models. We take advantage of a translational model property: $trans_r(\cdot)$ is a mapping from the head and relation to the tail in the vector space. Thus, given the query, the prototype can be determined by Lemma 1.

**Lemma 1.** *For an optimal translational model $\mathcal{T}$, the prototypes $P$ of $(h, r, ?)$ are:*

$$P = \{p | \mathrm{trans_r(emb(h))} = \mathrm{trans_r(emb(p))}\} \tag{5}$$

Lemma 1 employs the absolute equality of vectors for the ideal case. In practice, we relax this restriction by using the differentiable vector distance instead. In particular, we model the plausibility of a candidate prototype $p$ by:

$$f_{hr}(p) = \max(\gamma - \|\mathrm{trans_r(emb(h))} - \mathrm{trans_r(emb(p))}\|, 0) \tag{6}$$

where we use marginal distance because closer distance means higher plausibility. Symmetrically, we model the prototype for the query $(?, r, t)$ by:

$$f_{rt}(p) = \max(\gamma - \|\mathrm{trans_r^{-1}(emb(t))} - \mathrm{trans_r^{-1}(emb(p))}\|, 0) \tag{7}$$

where $trans_r^{-1}$ is the inverse of $trans_r$. For example, TransE [4] uses vector subtraction as $trans_r^{-1}$.

For a given query $(h, r, ?)$, we consider all candidate prototype entities whose relation $r$ is known in the training data. We aggregate these prototypes based on their scores in Eq. (6). The score of a tail entity $t$ is the sum of scores of its corresponding prototypes:

$$\mathcal{I}_{hr}(t) = 1/(\gamma |\{p | (p, r, t) \in \mathcal{D}\}|) \sum_{(p,r,t) \in \mathcal{D}} f_{hr}(p) \tag{8}$$

where $1/(\gamma |\{p | (p, r, t) \in \mathcal{D}\}|)$ is used to normalize the score. The score of a head entity $h$ for $(?, r, t)$ is computed symmetrically. We denote the above model $\mathcal{I}$ as IBLE (instance-based learning). We use the cross-entropy between $\mathcal{I}_{hr}(t)$ and the ground-truth entity as the training objective.

We highlight that IBLE's aggregating strategy in Eq. (8) differs from GNNs [23]. For query $(h, r, ?)$, regardless of whether the instance is a neighbor of $h$, we aggregate the instance $p$ throughout the full instance space whose relation $r$ is known (i.e. $\{p | (p, r, t) \in \mathcal{D}\}$).

### 3.4 Relation-awareness for Prototype Modeling

Next, using Eq. (6), we analyze the property of *relation-awareness* for prototype modeling. It is obvious that whether an entity is the prototype depends on the query relation. Joe Biden, for example, is a prototype of Jill Biden for lived_city, but not for profession. We summarize the relation-awareness of TransE, TransR and RotatE by substituting them into Eq. (6) in Table 1.

Table 1: Prototype modeling by translational models

| Model | Prototype modeling | Relation-aware |
|---|---|---|
| TransE | $\|\mathbf{e}_h - \mathbf{e}_p\|$ (10) | No |
| TransR | $\|\mathbf{W}_r\mathbf{e}_h - \mathbf{W}_r\mathbf{e}_p\|$ (11) | Yes |
| RotatE | $\|\mathbf{e}_h - \mathbf{e}_p\|$ (12) | No |
| R-RotatE | $\|\mathbf{W}_r\mathbf{e}_h - \mathbf{W}_r\mathbf{e}_p\|$ (13) | Yes |

Prototypes by TransE and RotatE are not relation-aware because Eq. (9) and Eq. (10) consider only entity embeddings and is agnostic to the relation $r$. The relational matrix $\mathbf{W}_r$, on the other hand, allows TransR to distinguish different relations.

Since RotatE is the cutting-edge translational model, we propose to optimize its representations. We integrate the relational matrix from TransR into RotatE, denoted as relational-RotatE (R-RotatE):

$$\text{R-RotatE}(h, r, t) = \|\mathbf{W}_r\mathbf{e}_h \circ r - \mathbf{W}_r\mathbf{e}_t\| \tag{14}$$

We show the prototype modeling of R-RotatE in Table 1.

**Remark 1.** *TransR and R-RotatE are relation-aware for prototype modeling, whereas TransE and RotatE are not.*

An interesting phenomenon is, TransE and RotatE actually have the same prototype model, as do TransR and R-RotatE. In this paper, we use the prototype modeling of TransR and R-RotatE by default.

# 4 Combine IBL and Translational Models

Embeddings of entities and relations of IBL can be directly reused in the translational model. In this section, we develop the theory of the joint optimization for IBL and the translational model. We show that their objectives are compatible. The theory motivates us to combine IBL with the translational model.

We denote the IBL model and the translational model with shared entity and relation embeddings $\theta$ as $\mathcal{I}_\theta$ and $\mathcal{T}_\theta$, respectively. We show that the optimization objectives of $\mathcal{I}^\theta$ and $\mathcal{T}^\theta$ are compatible in Lemma 2.

**Lemma 2.** *If we use margin loss for the translational model, and cross-entropy loss for the IBL model, and negative sampling to train both models, then the global minimum of the training criterion of $\mathcal{T}^\theta$ can be obtained only if $\mathcal{I}^\theta$ also reaches its optimum.*

This inspires us to propose CIBLE (combined instance-based learning) model, which combines $I_\theta$ and $T_\theta$ with the following scoring function:

$$\mathcal{C}_{\text{hr}}^\theta(\text{t}) = (1 - \alpha)\mathcal{I}_{\text{hr}}^\theta(\text{t}) + \frac{\alpha}{\gamma}\max(\gamma - \mathcal{T}^\theta(\text{h}, \text{r}, \text{t}), 0) \tag{15}$$

where $\alpha \in (0, 1)$ is a hyper-parameter. We use cross-entropy as the training objective for CIBLE. The optimization objectives of $\mathcal{T}^\theta$, $\mathcal{I}^\theta$ and $\mathcal{C}^\theta$ are compatible. We show this in Theorem 1

**Theorem 1.** *If we use cross-entropy loss for the combined model and the IBL model, the margin loss for the translational model, and train all three models with negative sampling, then the $\theta$ values for the global minimum training criteria of $\mathcal{C}_\theta$ and $\mathcal{T}_\theta$ are consistent. At that $\theta$, the training criterion of $\mathcal{I}_\theta$ is also minimized, and:*

$$\mathcal{C}_{\text{hr}}^\theta(\text{t}) = \begin{cases} 1 & (h, r, t) \text{ is a valid fact,} \\ 0 & \text{otherwise.} \end{cases} \tag{16}$$

# 5 Logical Rules as Prototype-based Inference

In this section, we show that, under the low-rank assumption, the instance-based equivalence and prototypes in IBL can be represented by a particular class of logical rules. We discovered that the interpretability of these rules differs from typical rules in previous papers. More numerical analysis will be presented in Sec 7.

Rule-based KBC models induct rules in the form of Horn clauses: $\text{rel}_1 \wedge \cdots \text{rel}_{\text{k}} \Rightarrow \text{rel}_0$. Table 2 shows examples from the body of previous publications [17, 38]. According to these examples, it is commonly believed that the *semantic compatibility* between the premise and the hypothesis provides the interpretability for humans. For example, the semantics of nationality is equal to the semantics of the path $\text{born\_in} \wedge \text{place\_in\_country}$, making the rule $\text{born\_in} \wedge \text{place\_in\_country} \Rightarrow \text{nationality}$ human-readable.

However, we found a special class of rules that challenge the understanding. Table 2 shows top-3 rules inducted by RNNLogic [17] for *profession* in FB15k-237. Clearly, these rules are in one of the following forms:

$$\text{rel}_1 \wedge \text{rel}_1^{-1} \wedge \text{rel}_0 \Rightarrow \text{rel}_0$$
$$\text{rel}_0 \wedge \text{rel}_1 \wedge \text{rel}_1^{-1} \Rightarrow \text{rel}_0 \tag{17}$$

The premises of both forms contain the composition of symmetric relations $\text{rel}_1 \wedge \text{rel}_1^{-1}$, whose meanings are in opposition to each other. The semantics of the premise alone cannot be understood by humans, making the connection between the premise and hypothesis unclear.

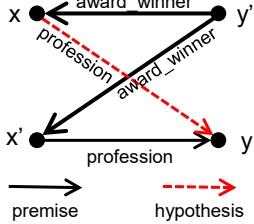

Figure 1: An IBL rule indicates the instance-based relational equivalence.

Table 2: Examples of rules.

**Top 3 rules for *profession* by RNNLogic [17].**
profession⇐award_winner$^{-1}$ ∧ award_winner ∧ profession
profession⇐nationality ∧ nationality$^{-1}$ ∧ profession
profession⇐webpage_category ∧ webpage_category$^{-1}$ ∧ profession

**Examples from the RNNLogic paper [17].**
nationality⇐born_in ∧ place_in_country
organization_in_state⇐organization_in_city ∧ city_locates_in_state

**Examples from the Neural LP paper [38].**
partially_contains⇐contains ∧ contains
marriage_location⇐nationality ∧ contains

However, after transforming such rules into the form of Fig. 1, their meanings become clearer. The semantics of such rules are consistent with IBL's low-rank assumption. The rule in Fig. 1 states that, conditional on the fact that $x$ and $x'$ are both the winners of award $y'$, they also tend to have the same profession $y$. As a result, the rule's true purpose is to find the prototype $x'$ to deduce the profession of $x$.

We employ translational models to provide a theoretical view of IBL rules. We take TransE as an example. In Theorem 2, we show that IBL rules always hold true under the assumption of translational models.

**Theorem 2** (The effectiveness of IBL rules). *For a KB with an optimal TransE model such that* $|e_h + r - e_t| = 0$ *iff* $(h, r, t) \in$ KB, $\forall r_0, r_1$, *the IBL rules* $rel_1 \wedge rel_1^{-1} \wedge rel_0 \Rightarrow rel_0$ *and* $rel_0 \wedge rel_1 \wedge rel_1^{-1} \Rightarrow rel_0$ *always hold.*

We highlight the difference in interpretability between the claims in previous papers and the rules in Eq. (17). We denote rules in Eq. (17) as *IBL rules*. IBL rules provide interpretability by establishing instance-based equivalence relations, whereas previous papers claimed that rules typically provide interpretability by semantic relevance between the premise and hypothesis. Experimental results show that the effect and interpretability of instance-based equivalence cannot be ignored. This is elaborated in Sec 7.

# 6 Experiments

## 6.1 Experimental Setup

**Datasets:** We select four typical KBC datasets for evaluation, including FB15k-237, WN18RR, Kinship, and UMLS [2]. For Kinship and UMLS, we use the training/validation/test division in [17]. Their statistics are shown in the Appendix.

**Baselines** We compared with the following types of KBC methods. *Knowledge graph embedding:* TransE [4], TransR [12], DistMult [37], ComplEx [30], ConvE [6], TuckER [2], and RotatE [27]. *Rule-based:* RNNLogic [17], NeuralLP [38], DRUM [21], PathRank [11], MINERVA [5], CTP [14] and M-Walk [24]. *Graph neural networks:* NBFNet [40], GraIL [29], and RGCN [23].

**Evaluation metrics** For each test triplet $(h, r, t)$, we construct two queries: $(h, r, ?)$ and $(?, r, t)$, with the answers $t$ and $h$. We choose Mean Rank (MR), Mean Reciprocal Rank (MRR), and hit@k under the filtered setting [27], which is consistent with most existing work.

**Implementation details** We follow [27] and employ negative sampling for training. All experiments can be run on a single Nvidia Tesla V100 GPU. We illustrate the hyper-parameter search process in the Appendix.

## 6.2 Main Results

We present the main results in Table 3 and Table 4. IBLE is competitive with most existing KBC methods. This verify the effect and potential of IBL, as the proposed IBLE is still primitive. In

---

[2]Available to researchers for non-commercial research and educational use.

Table 3: Knowledge base completion results on FB15k-237 and WN18RR. ‡ means the results are from [17], and † means the results are from [40].

| Category | Model | FB15k-237 | | | | | WN18RR | | | | |
|---|---|---|---|---|---|---|---|---|---|---|---|
| | | MR | MRR | H@1 | H@3 | H@10 | MR | MRR | H@1 | H@3 | H@10 |
| **KG Embedding** | TransE‡ | 357 | 0.294 | - | - | 46.5 | 3384 | 0.226 | - | - | 50.1 |
| | DistMult‡ | 254 | 0.241 | 15.5 | 26.3 | 41.9 | 5110 | 0.430 | 39.0 | 44.0 | 49.0 |
| | ComplEx‡ | 339 | 0.247 | 15.8 | 27.5 | 42.8 | 5261 | 0.440 | 41.0 | 46.0 | 51.0 |
| | ConvE‡ | 244 | 0.325 | 23.7 | 35.6 | 50.1 | 4187 | 0.430 | 40.0 | 44.0 | 52.0 |
| | TuckER‡ | - | 0.358 | 26.6 | 39.4 | 54.4 | - | 0.470 | 44.3 | 48.2 | 52.6 |
| | RotatE‡ | 177 | 0.338 | 24.1 | 37.5 | 53.3 | 3340 | 0.476 | 42.8 | 49.2 | 57.1 |
| **Rule-based Learning** | PathRank‡ | - | 0.087 | 7.4 | 9.2 | 11.2 | - | 0.189 | 17.1 | 20.0 | 22.5 |
| | M-Walk‡ | - | 0.232 | 16.5 | 24.3 | - | - | 0.437 | 41.4 | 44.5 | - |
| | NeuralLP‡ | - | 0.237 | 17.3 | 25.9 | 36.1 | - | 0.381 | 36.8 | 38.6 | 40.8 |
| | DRUM‡ | - | 0.238 | 17.4 | 26.1 | 36.4 | - | 0.382 | 36.9 | 38.8 | 41.0 |
| | RNNLogic‡ | 232 | 0.344 | 25.2 | 38.0 | 53.0 | 4615 | 0.483 | 44.6 | 49.7 | 55.8 |
| | RNNLogic+‡ | 178 | 0.349 | 25.8 | 38.5 | 53.3 | 4624 | 0.513 | 47.1 | 53.2 | 59.7 |
| **GNN-based Learning** | RGCN† | 221 | 0.273 | 18.2 | 30.3 | 45.6 | 2719 | 0.402 | 34.5 | 43.7 | 49.4 |
| | GraIL† | 2053 | - | - | - | - | 2539 | - | - | - | - |
| | NBFNet† | **114** | **0.415** | **32.1** | **45.4** | **59.9** | **636** | **0.551** | **49.7** | **57.3** | **66.6** |
| **IBL-based** | IBLE | 263 | 0.284 | 20.0 | 31.0 | 45.2 | 7205 | 0.394 | 37.7 | 40.0 | 42.7 |
| | CIBLE | 170 | 0.341 | 24.6 | 37.8 | 53.2 | 3400 | 0.490 | 44.6 | 50.7 | 57.5 |

Table 4: Knowledge base completion results on Kinship and UMLS.

| Category | Algorithm | Kinship | | | | | UMLS | | | | |
|---|---|---|---|---|---|---|---|---|---|---|---|
| | | MR | MRR | H@1 | H@3 | H@10 | MR | MRR | H@1 | H@3 | H@10 |
| **KG Embedding** | DistMult‡ | 8.5 | 0.354 | 18.9 | 40.0 | 75.5 | 14.6 | 0.391 | 25.6 | 44.5 | 66.9 |
| | ComplEx‡ | 7.8 | 0.418 | 24.2 | 49.9 | 81.2 | 13.6 | 0.411 | 27.3 | 46.8 | 70.0 |
| | ComplEx-N3‡ | - | 0.605 | 43.7 | 71.0 | 92.1 | - | 0.791 | 68.9 | 87.3 | 95.7 |
| | TuckER‡ | 6.2 | 0.603 | 46.2 | 69.8 | 86.3 | 5.7 | 0.732 | 62.5 | 81.2 | 90.9 |
| | RotatE‡ | 3.7 | 0.651 | 50.4 | 75.5 | 93.2 | 4.0 | 0.744 | 63.6 | 82.2 | 93.9 |
| **Rule Learning** | MLN‡ | 10.0 | 0.351 | 18.9 | 40.8 | 70.7 | 7.6 | 0.688 | 58.7 | 75.5 | 86.9 |
| | Boosted RDN‡ | 25.2 | 0.469 | 39.5 | 52.0 | 56.7 | 54.8 | 0.227 | 14.7 | 25.6 | 37.6 |
| | PathRank‡ | - | 0.369 | 27.2 | 41.6 | 67.3 | - | 0.197 | 14.8 | 21.4 | 25.2 |
| | NeuralLP‡ | 16.9 | 0.302 | 16.7 | 33.9 | 59.6 | 10.3 | 0.483 | 33.2 | 56.3 | 77.5 |
| | DRUM‡ | 11.6 | 0.334 | 18.3 | 37.8 | 67.5 | 8.4 | 0.548 | 35.8 | 69.9 | 85.4 |
| | MINERVA‡ | - | 0.401 | 23.5 | 46.7 | 76.6 | - | 0.564 | 42.6 | 65.8 | 81.4 |
| | CTP‡ | - | 0.335 | 17.7 | 37.6 | 70.3 | - | 0.404 | 28.8 | 43.0 | 67.4 |
| | RNNLogic‡ | 3.1 | 0.722 | 59.8 | 81.4 | 94.9 | 3.1 | 0.842 | 77.2 | 89.1 | 96.5 |
| **GNN-based** | NBFNet | 3.7 | 0.606 | 43.5 | 72.5 | 93.7 | 3.8 | 0.778 | 68.8 | 84.0 | 93.8 |
| **IBL-based** | IBLE | 3.7 | 0.650 | 51.3 | 75.5 | 93.7 | 3.2 | 0.816 | 71.7 | 90.0 | 96.1 |
| | CIBLE | **3.0** | **0.728** | **60.3** | **82.0** | **95.6** | **2.6** | **0.856** | **78.7** | **91.6** | **97.0** |

FB15k-237 and WN18RR, CIBLE outperforms previous state-of-the-art methods except the NBFNet. In Kinship and UMLS, CIBLE outperforms all existing methods. This verifies the effectiveness of CIBLE.

## 6.3 Instance-based Interpretations

**Understanding model behavior via prototypes** Table 5 shows the top 10 prototypes for predicting Taylor Swift and Barack Obama's nationalities in FB15k-237 according to Eq. (6). Despite the fact that they are both Americans, they have different prototypes. Taylor Swift prefers to use American singers and actors as prototypes, while Barack Obama prefers to use American politicians. These selections of prototypes are comprehensible to humans. In addition, we believe that prototypes are more transparent than logical rules, as many logical rules are semantically unclear to humans (see Table 2 for some examples). Sec 7 will delve deeper into this.

**Visualization of global model behavior** In CIBLE, the distance between entity embeddings (i.e. $trans_r(emb)$) measures the plausibility of a candidate prototype. Here we visualize

Table 5: Top prototypes. For the same relation (i.e. nationality), Taylor Swift and Barack Obama choose different prototypes. This illustrates the rationality of how the model infers.

| Entity & Relation | Prototypes |
|---|---|
| *Taylor Swift* /nationality | Carrie Underwood, Miley Cyrus, Selena Gomez, Taylor Lautner, Demi Lovato, Joe Jonas, Randy Travis, Garth Brooks, Brad Paisley, Trisha Yearwood |
| *Barack Obama* /nationality | Hillary Rodham Clinton, Al Gore, Jimmy Carter, Ossie Davis, James Madison, Martin Luther King, Jr., Paul Rudd, Daniel Inouye, Colin Powell, Herbert Hoover |

the entity embeddings (i.e. $\mathbf{W}_r\mathbf{e}$) for relation *official language* in FB15k-237 using t-SNE [31]. The visualization provides *global interpretation*, since it explains the prediction for all entities, rather than for one prediction.

The visualization provides rich information about how CIBLE makes predictions: (1) entities with the same value exhibit distinct clustering patterns, and (2) the distance between entities reflects knowledge relatedness beyond the target value. For example, *Argentina* (Spanish) and *Brazil* (Portuguese) are close to each other, since they share other relations . (3) The location interprets 1-to-N relations. For example, English, Mandarin, and Tamil are all official languages of Singapore, As a result, Singapore is close to China (Mandarin), India (English) and Sri Lanka (Tamil).

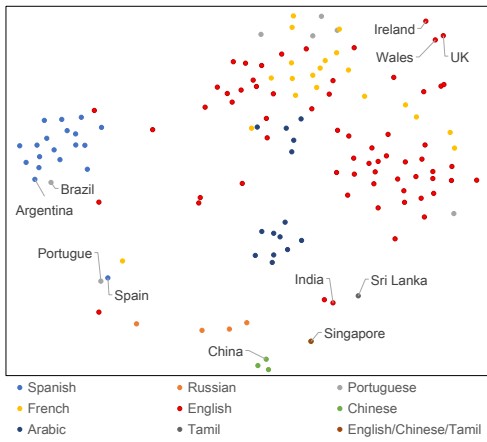

Figure 2: t-SNE visualization of entities on "/country/official_language" in FB15k-237.

## 6.4 Analysis

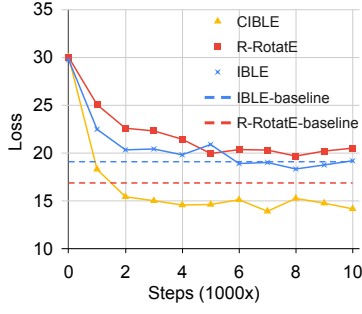

Figure 3: Training loss on UMLS.

Table 6: Effect of relation-awareness.

| Model | Relation -aware | FB15k-237 | | WN18RR | |
|---|---|---|---|---|---|
| | | MRR | H@10 | MRR | H@10 |
| CIBLE-TransE | No | 0.286 | 44.7 | 0.236 | **53.8** |
| CIBLE-TransR | Yes | **0.341** | **52.2** | **0.250** | 49.7 |
| CIBLE-RotatE | No | 0.315 | 50.6 | 0.430 | 54.2 |
| CIBLE-R-RotatE | Yes | **0.341** | **53.2** | **0.490** | **57.5** |

**Consistency of model optimization** In Theorem 1, we show the consistency of the optimization objectives for the three models. To verify this theory, we show their training losses when optimizing CIBLE in Fig. 3. IBLE-baseline and RotatE-baseline denote the convergence of the training loss when we optimize them separately. Overall, as we optimizing CIBLE, the training loss of IBLE and RotatE both decreases. In addition, optimizing CIBLE helps IBLE converge to its minimum training loss. This verifies the consistency of the three optimization objectives.

**Effect of relation-awareness** We show in Sec 3.4 that prototypes modeled by TransR and R-RotatE are relation-aware while TransE and RotatE are not. To validate the effect of relation-awareness, we compare their effect when adapting to CIBLE. Table 6 shows the results. Models with relation-awareness perform better on most metrics. This is in line with our intuition about prototypes.

## 7 Rule-based Reasoners: Semantic Relevance or Instance-based Equivalence?

In Sec 5, we defined a special class of logical rules, namely IBL rules. In this section, we reassess the mechanism and interpretability of logical rules by validating the effect of IBL rules.

Table 7: Effect of IBL rules for FB15k-237 and WN18RR.

| Model | FB15k-237 | | | | | WN18RR | | | | |
|---|---|---|---|---|---|---|---|---|---|---|
| | MR | MRR | H@1 | H@3 | H@10 | MR | MRR | H@1 | H@3 | H@10 |
| all rules | 178 | 0.349 | 25.8 | 38.5 | 53.3 | 4624 | 51.3 | 47.1 | 53.2 | 59.7 |
| non-IBL rules only | 1241 | 0.317 | 23.2 | 35.1 | 48.6 | 9849 | 44.1 | 41.9 | 45.3 | 48.5 |
| IBL-rules only | 1011 | 0.326 | 24.0 | 36.1 | 49.9 | 8946 | 44.0 | 40.6 | 46.3 | 49.9 |

Table 8: Effect of IBL rules for Kinship and UMLS.

| Model | Kinship | | | | | UMLS | | | | |
|---|---|---|---|---|---|---|---|---|---|---|
| | MR | MRR | H@1 | H@3 | H@10 | MR | MRR | H@1 | H@3 | H@10 |
| all rules | 3.1 | 0.842 | 77.2 | 89.1 | 96.5 | 3.1 | 0.722 | 59.8 | 81.4 | 94.9 |
| non-IBL rules only | 9.9 | 0.667 | 53.7 | 76.9 | 86.2 | 4.0 | 0.681 | 55.0 | 77.3 | 93.2 |
| IBL-rules only | 3.0 | 0.840 | 75.1 | 91.3 | 96.6 | 3.3 | 72.1 | 59.5 | 81.4 | 94.7 |

Earlier rule-based models learn logical rules in a differential way [21, 38]. However, their search space is exponentially large, which results in limited rule length. The recently proposed method, RNNLogic [17], was a breakthrough in its capacity to generate rules with maximum length of 5. With these longer rules, RNNLogic achieves 53.3 hit@10 in FB15k-237, compared to the previous highest hit@10 of only 36.4 for rule-based models [21]. Therefore, we use RNNLogic to investigate the effect of IBL rules.

To demonstrate the effect of IBL rules, we control the candidate rule space of RNNLogic by modifying its rule collection module. We carried out the following ablations: (1) collecting rules from the entire rule space; (2) collecting only non-IBL rules; (3) collecting only IBL rules. Table 7 and Table 8 show the effects of these ablations.

The results are surprising. Despite the fact that IBL rules only cover a small portion of the entire rule space, learning IBL rules outperforms learning non-IBL rules in all four benchmarks. In Kinship and UMLS, learning only IBL rules has almost no performance degradation when compared to learning from the entire rule space. The results indicate that, the small number of IBL rules are even more critical than non-IBL rules.

These findings challenge the widely-held belief that rule-based reasoning use rules whose premises and hypotheses are semantically compatible. The results imply that rule-based KBC models work largely because its capability to represent instance-based equivalence (i.e. prototypes) via IBL rules. The interpretability from instance-based equivalency should not be overlooked in rule-based models.

To understand why IBL rules outperform other semantic non-IBL rules, we investigate the quality of each rule. More concretely, we show the average precision and support [7] of each collected rule for different rule types in Table 9. The results show that both the average precision and support of IBL rules are substantially higher than those of non-IBL rules. The high quality of IBL rules explains why they outperform non-IBL rules despite occupying only a small fraction of the entire rule space.

**IBL-based vs rule-based models** When representing the instance-based equivalence relationship, CIBLE is more flexible and theoretically sound than rule-based models using IBL rules. In Table 3 4, despite the fact that RNNLogic employs a complex EM algorithm for learning, CIBLE outperforms RNNLogic in three of the four benchmark. Therefore, we consider CIBLE to be a more effective alternative to the rule-based models in instance-based equivalency modeling.

Table 9: Average support and precision of IBL and non-IBLE rules.

| | FB15k237 | WN18RR | UMLS | Kinships |
|---|---|---|---|---|
| IBL Rule | **708.3 / 3.7%** | **2374.3 / 12.7%** | **3.0 / 11.6%** | **8.7 / 11.6%** |
| Non-IBL Rule | 281.4 / 1.7% | 188.3 / 5.0% | 3.0 / 9.5% | 6.7 / 5.1% |

## 8 Conclusion

In this paper, we explore a novel IBL-based method for KBC. We validate its effectiveness in two aspects. First, we show that the prototypes can be expressed in a closed-form with the well-studied translational models, and IBL-based models can be cotypeombined with translational models. Second, we found that rule-based reasoning relies heavily on IBL rules. This challenges previous common understanding of how logical rules work and provide interpretability.

**Broader impact** Our proposed IBL method provides a new paradigm for learning representations for knowledge bases. Based on the results of the experiments, the IBL paradigm can be combined with translational models and rule-based reasoners and enhance the effectiveness. Our theory has provided the foundation for combing IBL models and translational models. We expect to better exploit the theory to combine different KBC models in the future.

## Acknowledgments and Disclosure of Funding

This paper was supported by National Natural Science Foundation of China (No. 61906116), by Shanghai Sailing Program (No. 19YF1414700). We thank Wenting Ba for her valuable plotting assistance.

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
