## A  Proof of Lemma 1

*Proof.* In an optimal translational model $\mathcal{T}$, for any valid $(h, r, t)$ we have:
$$\text{trans}_r(\text{emb}(h)) = \text{emb}(t) \tag{18}$$
According to the definition of prototypes, an entity $p$ is the prototype for $(h, r, ?)$ if and only if:
$$\text{trans}_r(\text{emb}(h)) = \text{trans}_r(\text{emb}(p)) \tag{19}$$
$\square$

## B  Proof of Lemma 2

*Proof.* Recall that the margin loss of $\mathcal{T}^\theta$ with margin $\gamma$ is:
$$\mathcal{L} = -\max(\gamma - \mathcal{T}^\theta(h, r, t), 0) + \sum_{i=1}^n \frac{1}{n} \max(\gamma - \mathcal{T}^\theta(h_i', r, t_i'), 0) \tag{20}$$
where $(h_i', r, t_i')$ is the $i$-th negative triplet.

If the global minimum of the loss is achieved, then for any positive $(h, r, t)$, we have:
$$\max(\gamma - \mathcal{T}^\theta_{hr}(t), 0) = \gamma \Rightarrow \mathcal{T}^\theta_{hr}(t) = 0 \tag{21}$$
for any negative $(h, r, t_{neg})$, we have:
$$\max(\gamma - \mathcal{T}^\theta_{hr}(t_{neg}), 0) = 0 \Rightarrow \mathcal{T}^\theta_{hr}(t_{neg}) \geq \gamma \tag{22}$$

Then for the positive prototype $p$ of $(h, r, t)$, we have:
$$f_{hr}(p) = \gamma \tag{23}$$
For the negative prototype $p_{neg}$ of $(h, r, t)$, we have:
$$f_{hr}(p_{neg}) = 0 \tag{24}$$

With Eq. (8), the score of an candidate tail $t'$ is:
$$\mathcal{I}^\theta_{hr}(t') = \begin{cases} 1 & (h, r, t') \text{ is positive} \\ 0 & \text{otherwise.} \end{cases} \tag{25}$$

And the cross-entropy loss for $\mathcal{I}^\theta$ is minimized. $\square$

## C  Proof of Theorem 2

*Proof.* In an optimal TransE model, for all entities $a, b, c, d$ that satisfy the premise of the IBL rule (i.e. $(a, r_0, b), (b, r_1, c), (c, r_1^{-1}, d) \in KB$), we have
$$\|e_a + r_0 - e_b\| = 0, \|e_b + r_1 - e_c\| = 0, \|e_c + r_1^{-1} - e_d\| = 0$$

Therefore, $e_d = e_a + r_0$, $\|e_a + r_0 - e_d\| = 0$.

As a result, $(a, r_0, d) \in KB$, which indicates that the hypothesis of the IBL rule also holds. So the IBL rule $r_0 \wedge r_1 \wedge r_1^{-1} \implies r_0$ always holds. $\square$

## D  Hyperparameters

We search hyperparameters from the following range: learning rate $l \in \{1 \times 10^{-5}, 2 \times 10^{-5}, 5 \times 10^{-5}, 1 \times 10^{-4}, 2 \times 10^{-4}, 5 \times 10^{-4}\}$, batch size $b \in \{8, 16, 32, 64, 128, 256, 512, 1024\}$, dimension of embedding $d \in \{200, 500, 1000, 2000\}$, and margin $\gamma \in \{3, 6, 9, 12, 15, 18\}$. We use wandb [3] to search for best hyperparameters.

## E  Dataset Statistics

We summarize the number of entities, relations and examples in each split for four benchmarks in our experiments in Table 10.

---

[3] https://wandb.ai/home

Table 10: Dataset statistics.

| Dataset | #Entities | #Relations | #Train | #Validation | #Test |
|---------|-----------|------------|--------|-------------|-------|
| FB15k-237 | 14,541 | 237 | 272,115 | 17,535 | 20,466 |
| WN18RR | 40,943 | 11 | 86,835 | 3,034 | 3,134 |
| Kinship | 104 | 25 | 3,206 | 2,137 | 5,343 |
| UMLS | 135 | 46 | 1,959 | 1,306 | 3,264 |