# OpenReview forum: "Instance-based Learning for Knowledge Base Completion"
_NeurIPS.cc/2022/Conference — NeurIPS 2022 Accept_

### Official Review · Reviewer_DVJo · 2022-07-09

**Rating:** 4
**Confidence:** 3
**Soundness:** 2 fair
**Presentation:** 2 fair
**Contribution:** 2 fair

**Summary:**

The paper studies the knowledge base completion problem and proposes an instance-based learning method that utilizes the prototypes to help the inference of a given query. Then the relation information can be used in the prototype modeling and could obtain better performance. The proposed IBL can be combined with the traditional translation models to achieve the best performance. The authors also compare their model with other existing methods of rule discovery. IBL role provides interpretability by establishing instance-based equivalence relations.

**Questions:**

- The proposed instance-based learning looks like the neighboring entities of a relation can be aggregated to help infer the missing facts. Can you please discuss the differences between the proposed and GNN-based methods, e.g., R-GCN?
- The motivation for combining IBL and Translational models is not clear. The authors mention the objectives are compatible. However, they did not clarify why the IBL cannot work well alone and have to work with the translation models.
- It is not easy to understand why the IBL rule provides better interpretability than existing methods. I wonder why the IBL using instance-based equivalence relations can work better than the methods adopting semantic relevance.

**Limitations:**

Please discuss the limitations and negative societal impact of the work.

**Strengths And Weaknesses:**

Strengths:
- Most important baselines are included in the experiments.
- The case study and visualization are interesting.
- The conducted experiments are comprehensive.

Weaknesses:
- The proposed instance-based method needs to work with the traditional translation models to obtain the best performance. It looks like a neighborhood aggregation to enhance the translation models as they do not consider the graph structure information. Based on that, the paper provides an incremental strategy to improve the existing methods.
- The presentation about the logical rule is not clear and not easy to follow.
- The motivation for combining IBL and Translational models is not clarified.

---

> ### Author Response · Authors · 2022-08-02
> **Response to Reviewer DVJo**
>
> We thank the reviewer for the feedback and respond to their concerns below.
> >**Q1: The proposed instance-based learning looks like the neighboring entities of a relation can be aggregated to help infer the missing facts. Can you please discuss the differences between the proposed and GNN-based methods,e.g., R-GCN? It looks like a neighborhood aggregation to enhance the translation models.**
>
> This misunderstands how our model works. Our model's aggregating strategy differs significantly from GNNs. In Eq. (8), for query $(h,r,?)$, regardless of whether the instance is a neighbor of $h$, we **aggregate the instances throughout the full instance space** whose relation $r$ is known. The strategy follows the common setting of instance-based learning. A GNN like R-GCN, on the other hand, only **aggregates $h$'s neighbors**. As a result, neither a neighborhood-enhanced translation model nor a GNN model should be used to describe the proposed model.
>
> >**Q2: The presentation about the logical rule is not clear and not easy to follow.**
>
> Definition: IBL rules are rules in the form of either
>
> $rel_1 \land rel^{-1}_1 \land rel_0 \Rightarrow rel_0$
>
> or
>
> $rel_0 \land rel_1 \land rel^{−1}_1 \Rightarrow rel_0$.
>
> The premises of both forms contain symmetric relations $rel_1$ and $rel^{-1}_1$, whose meanings are in opposition to each other (line 187-189).
> Unlike previous rules that establish the semantic relevance between the premise and the hypothesis, IBL rules actually establish the instance-based equivalence relations (line 193-196).
>
> >**Q3: They did not clarify why the IBL cannot work well alone and have to work with the translation models.**
>
> It should be noted that IBL alone (IBLE) only employs the marginal distance and summation (Eq. (6)(7)(8)). Despite its simplicity, IBLE still outperforms all other interpretable rule-based learning models, with the exception of RNNLogic, which is far more complex and depends on an EM workflow and the PNA aggregator. The simple structure also leads to greater interpretability (e.g., the global model behavior provided in Fig. 2).
>
> In fact, the prototyping in Eq. (6)(7) can be directly enhanced by more complicated models. For instance, using GNNs, we can enhance the prototyping by adding the path information between $h$ and $p$. This, however, is not the focus of this paper. We believe that a simpler solution better displays the mechanism and effect of instance-based learning for KB completion.
>
> > **Q4: It is not easy to understand why the IBL rule provides better interpretability than existing methods. I wonder why the IBL rule using instance-based equivalence relations can work better than the methods adopting semantic relevance.**
>
> This is a great point. To understand why IBL rules outperform other semantic relevance-based rules (non-IBL rules), we investigate the quality of each rule. More concretely, we show the average precision and support [1] of each collected rule for different rule types below.
>
> |              |   FB15k-237   |           |    WN18RR    |           |     UMLS     |           |    Kinship   |           |
> |--------------|:------------:|:---------:|:------------:|:---------:|:------------:|:---------:|:------------:|-----------|
> |              | support | prec. | support | prec. | support | prec. | support | prec. |
> | IBL Rule     |    **708.26**    |    **3.74%**   |    **2374.28**   |    **12.7%**   |     **3.04**     |   **11.64%**   |     **8.65**     |   **11.58%**   |
> | Non-IBL Rule |    281.36    |    1.70%    |    188.29    |    4.92%   |     2.99     |    9.52%   |     6.71     |    5.09%   |
>
> The results show that both the average precision and support of IBL rules are substantially higher than those of non-IBL rules. The support of IBL rules in WN18RR is even about one order of magnitude higher. The findings explain why IBL rules outperform non-IBL rules despite occupying only a small fraction of the entire rule space (Table 7, 8).
>
> [1]Galárraga, L., Teflioudi, C., Hose, K., & Suchanek, F. M. (2015). Fast rule mining in ontological knowledge bases with AMIE+. The VLDB Journal, 24(6), 707-730.

---

> > ### Author Response · Authors · 2022-08-07
> > **A further intuitive explanation of Q4**
> >
> > We would like to further explain the **intuition** about why IBL rules are of higher quality than non-IBL rules, especially in terms of support. This is due to the fact that the real KBs satisfy the **low-rank assumption** in the instance-based setting (line 39-43).
> >
> > More specifically, as stated in line 191-194, the IBL rule actually describes that "if $x$ and $x'$ have the same $rel_1$, then they tend to have the same $rel_0$". An IBL rule holds whenever $x$ and $x'$ have two relations ($rel_0$, $rel_1$) with the same values.  Real KBs often have a large number of entities with identical relations and values. Continuing with the example in Fig. 1, the first rule means that two entities that win the same awards also tend to have the same professions. In FB15k-237, we found 100 entities with the profession of `actor` and won the `Screen Actors Guild Award`. As a result, it is easy for the IBL rule to hold in real KBs due to the low-rank assumption.

---

> > ### Comment · Reviewer_DVJo · 2022-08-09
> > **Thanks for the response**
> >
> > Thanks to the authors for the response. The response addresses a part of my concerns. However, I am still not convinced by the answer for Q3. And authors are encouraged to provide theoretical proof to explain the Q4 as the empirical results cannot help a lot on explanation. After reading the response and other reviews, I tend to keep my score.

---

> > > ### Author Response · Authors · 2022-08-09
> > > **Thanks for your suggestion of providing a theoretical proof for the effect of IBL rules (about Q4)**
> > >
> > > We employ translational models to provide a theoretical proof for the effectiveness of IBL rules. We will show that IBL rules always hold true under the assumption of translational models. We take TransE as an example. Consider an optimal TransE model such that $\lVert e_h + r - e_t \rVert= 0$ iff $ (h,r,t) \in KB$.
> > >
> > > *(The effectiveness of IBL rules)* $\forall r_0, r_1$, the IBL rule $r_0 \land r_1 \land r_1^{-1} \Rightarrow r_0$ always holds.
> > >
> > > **Proof:** For all entities $a$, $b$, $c$, and $d$ that satisfy the premise of the IBL rule (i.e., $(a, r_0, b), (b, r_1, c), (c, r_1^{-1}, d) \in KB$) , we have:
> > >
> > > $\lVert e_a + r_0 - e_b \rVert = 0, \lVert e_b + r_1 - e_c \rVert = 0, \lVert e_c + r_1^{-1} - e_d \rVert = 0 $,
> > >
> > > Therefore, $e_d = e_a + r_0，\lVert e_a + r_0 - e_d \rVert = 0$
> > >
> > > As a result,  $(a, r_0, d) \in KB$, which indicates that the hypothesis of the IBL rule also holds. So the IBL rule $r_0 \land r_1 \land r_1^{-1} \Rightarrow r_0$ always holds.
> > >
> > > Obviously, the above proof can be generalized to other translational models which represent inversion and compositionality (e.g., TransR, RotatE). On the other hand, semantic relevance-based rules (non-IBL rules) do not always hold (e.g., $r_0 \Rightarrow r_1$). The above proof demonstrates the effect of the IBL rule and explains why IBL rules have higher empirical support/precision. This is a great point. We will include it in the revised version.

---

> ### Author Response · Authors · 2022-08-09
> **We are looking forward to continuing to communicate with you.**
>
> Dear Reviewer DVJo,
>
> We sincerely thank you for acknowledging that the case study and visualization are interesting, and IBL rules provide new interpretability by establishing instance-based equivalence relations.
>
> We have also carefully considered your questions and suggestions and given our responses. **We are looking forward to continuing to communicate with you. If you have any further questions, please let us know.** Thank you.

---

### Official Review · Reviewer_ZosV · 2022-07-11

**Rating:** 7
**Confidence:** 3
**Soundness:** 4 excellent
**Presentation:** 3 good
**Contribution:** 3 good

**Summary:**

This paper proposes Instance-Based Learning for translational embeddings, and evaluates its performance in knowledge-base construction tasks. The paper first defines the general concept of prototyping in translational embedding training using relational tuples, then discuss methods to train with both translational loss and IBL loss, which is through a linear combination of their respective loss functions controlled using a hyper-parameter weight alpha. The paper also presents experimental results on 4 KBC datasets, which largely surpass performances of all but 1 graph network based approach, and does so with much less computational complexity. The paper also interestingly present the finding that it seems like rule-based reasoning doesn't necessarily always find semantically relevant rules.

**Questions:**

1. One could probably argue, that prototyping is potentially a GNN approach with limited depth (1). Recent work on GNN for KBC have considerable difficulty with computational costs, which led to a series of path-based learning algorithms. Based on the simplicity and strong performance of CIBLE, I wonder if it could be extended to a GNN with greater depth?

2. Prototyping like in CIBLE does have advantages over plain TransR and RotatE as demonstrated in this paper. However, I couldn't figure out why there seems to be quite a gap between the performance of TransE/IBLE against CIBLE, but not so much between TransE and IBLE. I wish the authors could maybe explain a bit on why this is the case. It would be especially helpful if the authors could include their own implementation of TransR's evaluation on the table if not TransE/R-RotatE as well, since as I understand CIBLE is the combination of IBLE loss and TransR, this should not be very difficult.

**Limitations:**

I do not see potential negative societal impact.

**Strengths And Weaknesses:**

Strength:
1. The paper is clearly written and presented. The mathematically formulation is well done and easy to understand.
2. The paper proposes a method with relatively low-computational cost (compared to graph neural networks) to complement existing translational embedding learning method.
3. The paper shows that the proposed method is highly competitive in all experimented datasets.

Weakness:
I could not find major weaknesses in this paper. I think it would be better to include numbers of the authors' implementation of TransE, TransR, etc. (without IBE loss) in the evaluation table as an ablation study perhaps. But all in all, IBLE itself is already an interesting approach.

---

> ### Author Response · Authors · 2022-08-02
> **Response to Reviewer ZosV**
>
> We appreciate the thoughtful and positive feedback. Please see our comments below.
> >**Q1: One could probably argue that prototyping is potentially a GNN approach with limited depth. I wonder if it could be extended to a GNN with greater depth?**
>
> Our model's aggregating strategy differs significantly from GNNs. In Eq. (8), for query $(h,r,?)$, regardless of whether the instance is a neighbor of $h$, we **aggregate the instances throughout the full instance space** whose relation $r$ is known. The strategy follows the common setting of instance-based learning. A GNN like R-GCN, on the other hand, only **aggregates $h$'s neighbors**. Therefore, prototyping is not a GNN with limited depth.
>
> In addition, it is possible to enhance prototyping using GNNs. One promising way is to use the path information between two entities (represented by GNNs) to improve the prototyping in Eq. (6)(7). This enhancement should not be described as a progression from shallower GNNs to deeper GNNs.
>
> >**Q2: Prototyping like in CIBLE does have advantages over plain TransR and RotatE as demonstrated in this paper. However, I couldn't figure out why there seems to be quite a gap between the performance of TransE/IBLE against CIBLE, but not so much between TransE and IBLE. It would be especially helpful if the authors could include their own implementation of TransR's evaluation on the table if not TransE/R-RotatE as well. It would be better to include numbers of the authors' implementation of TransE, TransR, etc. (without IBE loss) in the evaluation table as an ablation study perhaps.**
>
> We show the results of our implementation of TransE/TransR and CIBLE built on top of them below. We use the same hyper-parameter search space for all experiments.
>
> |                | FB15k-237 |        | WN18RR |        |
> |----------------|----------|--------|--------|--------|
> |                | MRR      | Hits@10 | MRR    | Hits@10 |
> | TransE (ours)  |   0.237  |  41.1  |  0.208 |  49.5  |
> | CIBLE-TransE   |   **0.286**  |  **44.7**  |  **0.236** |  **53.8**  |
> | TransR (ours)  |   0.211  |  39.2  |  0.193 |  45.9  |
> | CIBLE-TransR |   **0.341**  |  **52.2**  |  **0.250**  |  **49.7**  |
> | RotatE (ours)   |   0.338  |  53.3  |  0.476 |  57.1  |
> | CIBLE-RotatE |   **0.341**  |  **53.7**  |  **0.490**  |  **57.5**  |
>
> It can be seen that the CIBLE model consistently outperforms the translational models. This validates the effectiveness of the model and theory of CIBLE.

---

> ### Author Response · Authors · 2022-08-09
> **We are looking forward to continuing to communicate with you.**
>
> Dear Reviewer ZosV,
>
> We sincerely appreciate your acknowledging that the mathematical formulation in the paper is well-presented, the method is effective despite its computational simplicity, and the finding of how rule-based reasoning works is interesting.
>
> We have also carefully considered your questions and suggestions and given our responses. **We are looking forward to continuing to communicate with you. If you have any further questions, please let us know.** Thank you.

---

### Official Review · Reviewer_qiwG · 2022-07-11

**Rating:** 6
**Confidence:** 3
**Soundness:** 3 good
**Presentation:** 3 good
**Contribution:** 2 fair

**Summary:**

This paper introduces a new paradigm for KBC task by exploring instance based learning. They show that most of the
rule learning models mostly learn instance-based rules and using only those rules they can still perform close to state of the art.
Paper also proposes ways to finding prototypes and propose models on how IBL can be combined with other approaches.
Experimental results show that its competent on several benchmarks though it doesn't outperform existing systems.
Using existing embedding methods to select prototype seems novel and builds on top of the existing literature.


Any comments on the practical scenarios with large KBs and long skewed entities and relationships? Is it
is to find a prototype entity to answer the queries for relatively less represented entity types and relationships?



**Questions:**

Please see summary

**Limitations:**

Yes

**Strengths And Weaknesses:**

Strengths:
Bringing IBL to KBC space and see how it performs with respect to current systems

weaknesses:
Experimental results are not on par with other benchmarks.
Not sure about how it can extend to practical use cases with long tail entities.

---

> ### Author Response · Authors · 2022-08-02
> **Response to Reviewer qiwG**
>
> We thank the reviewer for the positive feedback. Please see our responses below.
> >**Q1: Any comments on the practical scenarios with large KBs and long skewed entities and relationships? Is it is to find a prototype entity to answer the queries for relatively less represented entity types and relationships?**
>
> Yes, for long skewed entities and relationships, IBLE makes predictions by finding prototypes.
>
> To validate the effect, we report the performance of our model for long-tailed entities and relations. The table below shows the results for the rarest 10% of entities and relationships. Our model outperforms baselines in most cases. We believe that this is due to the fact that instance-based learning is easier in the few-shot setting [1][2].
>
> **Performance on Top 10% rarest entities**
> |        | FB15k-237 |        | WN18RR |        | UMLS  |        | Kinship |        |
> |--------|----------|--------|--------|--------|-------|--------|---------|--------|
> |        | MRR      | Hits@10 | MRR    | Hits@10 | MRR   | Hits@10 | MRR     | Hits@10 |
> | RotatE |    0.095 |   14.2 |  **0.074** |   11.5 | 0.239 |   37.7 |    0.590 |   91.9 |
> | TransE |    0.088 |   **15.9** |   0.040 |   10.3 | 0.222 |   **55.7** |   0.222 |   69.8 |
> | CIBLE  |    **0.105** |   **15.9** |  0.067 |   **11.7** | **0.434** |   51.6 |   **0.684** |   **94.7** |
>
> **Performance on Top 10% rarest relations**
> |        | FB15k-237  |          | WN18RR    |          | UMLS      |        | Kinship   |          |
> |--------|-----------|----------|-----------|----------|-----------|--------|-----------|----------|
> |        | MRR       | Hits@10   | MRR       | Hits@10   | MRR       | Hits@10 | MRR       | Hits@10   |
> | RotatE | **0.522** | **68.4** |      0.35 |     40.7 |     0.587 |   72.6 |     0.404 |     68.8 |
> | TransE |     0.474 |    0.631 |     0.133 |     33.3 |     0.361 |   58.9 |     0.234 |       56 |
> | CIBLE  |     0.514 |     68.1 | **0.431** | **53.7** | **0.611** | **75** | **0.477** | **78.4** |
>
> Besides, another key point for the practical scenario is that KBs rapidly grow. Therefore it is crucial for the model to make predictions for unseen/new knowledge. And as we introduced in lines 37-39, instance-based learning ensures high-quality reasoning as the KB grows dynamically. For example, when predicting Jill Biden's lived city, even if Jill Biden moves to another city (possibly a new entity) in the future, we can still use Joe Biden as the prototype to make predictions.
>
> [1] Vinyals, Oriol, et al. "Matching networks for one-shot learning." Advances in neural information processing systems 29 (2016).
>
> [2] Snell, Jake, et al. "Prototypical networks for few-shot learning." Advances in neural information processing systems 30 (2017).

---

> > ### Comment · Reviewer_qiwG · 2022-08-09
> > **response**
> >
> > Thanks for providing answers to my questions.

---

> > > ### Author Response · Authors · 2022-08-09
> > > **We thank you for acknowledging our responses and raising the score.**
> > >
> > > We are excited to hear that our responses answered your questions. Thank you.

---

> ### Author Response · Authors · 2022-08-09
> **We are looking forward to continuing to communicate with you.**
>
> Dear Reviewer qiwG,
>
> We sincerely appreciate your acknowledging that we proposed a new paradigm for KBC and that using embedding methods to select prototypes is novel.
>
> We have also carefully considered your questions and suggestions and given our responses. **We are looking forward to continuing to communicate with you. If you have any further questions, please let us know.** Thank you.

---

### Official Review · Reviewer_iZrV · 2022-07-12

**Rating:** 5
**Confidence:** 4
**Soundness:** 2 fair
**Presentation:** 3 good
**Contribution:** 3 good

**Summary:**

This work proposed to use prototype entities to enhance the knowledge base completion methods. Specifically, the KBC methods firstly look for the alternative entities that share the same value with the query entity and relation as the prototype entities. The authors show that the prototypes have closed-form expressions when combining with translation-based KBC models like TransE. The experimental results show it outperforms existing methods. Besides, this work also claims the connections between instance-based learning and rule-based learning. Further it encourages the rethinking of interpretability of rule-based reasoning from the angle of instance-based equivalence.


**Questions:**

1. Can you demonstrate more examples of IBL rules obtained from the proposed methods? It could be better to have detailed comparisons with the rules from previous work.

2. The assumption is a bit in line with collaborative filtering in the recommendations. Have you thought of any solutions from the perspective of matrix factorization?

3. Can the prototype-based methods generalize to unseen relations or OOD scenarios?


**Limitations:**

N.A

**Strengths And Weaknesses:**

Strengths:
1. The idea is well-motivated and prototype-based methods have been shown useful for many downstream tasks such as relation extraction etc. It is good to apply to knowledge base completion tasks.

2. The closed-form solution of looking for prototype entities is interesting under the assumption of translation-based models.

3. The proposed instance-based learning method can achieve competitive performance and then combining with translation-based methods outperforms the state-of-the-art methods.


Weakness:
1. The introduction of logical rules as prototype-based inference is vague. One concrete example in Section 5 is not convincing and clearly states the connections or significant differences between prototype-based inference and rule-based reasoning.
2. The IBL rules should be formally defined in Section 5.
3. The claim that replacing the rule-based reasoning with instance-based learning is a bit weak from the limited experimental results.

---

> ### Author Response · Authors · 2022-08-02
> **Response to Reviewer iZrV**
>
> We thank the reviewer for the insightful and positive feedback. Please see our responses below.
> > **Q1: The IBL rules should be formally defined in Section 5. One concrete example in Section 5 is not convincing and clearly states the connections or significant differences between prototype-based inference and rule-based reasoning.**
>
> Definition: IBL rules are rules in the form of either
>
> $rel_1 \land rel^{-1}_1 \land rel_0 \Rightarrow rel_0$
>
> or
>
> $rel_0 \land rel_1 \land rel^{−1}_1 \Rightarrow rel_0$.
>
> The premises of both forms contain a pair of symmetric relations $rel_1$ and $rel^{-1}_1$, whose meanings are in opposition to each other (line 187-189).
>
> As illustrated in Fig. 1, IBL rules actually represent the instance-based equivalence (line 193-196). To help reviewers understand the difference between IBL rules and typical rules in previous papers which represent semantic relevance, we added explanations for each rule in Table 2 below.
>
> **Top 3 rules for *profession* in FB15k-237 by RNNLogic**
> ***
> **(IBL rule)** *award_winner*$^{−1}$ $\land$ *award_winner* $\land$ *profession* $\Rightarrow$ *profession*.
>
> Explanation: If $x$ and $x'$ both win award $y'$, then they tend to have the same profession $y$.
> ***
> **(IBL rule)** *nationality* $\land$ *nationality*$^{−1}$ $\land$ *profession* $\Rightarrow$ *profession*.
>
> Explanation: If $x$ and $x'$ both have the nationality $y'$, then they tend to have the same profession $y$.
> ***
> **(IBL rule)** *webpage_category* $\land$ *webpage_category*$^{−1}$ $\land$ *profession* $\Rightarrow$ *profession*.
>
> Explanation: If $x$ and $x'$ both have the webpage category $y'$, then they tend to have the same profession $y$.
> ***
>
> **Examples from the RNNLogic paper**
> ***
> **(non-IBL rule)** *born_in* $\land$ *place_in_country* $\Rightarrow$ *nationality*.
>
> Explanation: If $x$ was born in place $z$, and $z$ is in country $y$, then $x$ tends to have nationality $y$.
> ***
> **(non-IBL rule)** *organization_in_city* $\land$ *city_locates_in_state* $\Rightarrow$ *organization_in_state*.
>
> Explanation: If $x$ is in city $z$, and $z$ locates in state $y$, then $x$ tends to locate in state $y$.
> ***
>
> > **Q2: The claim that replacing the rule-based reasoning with instance-based learning is a bit weak from the limited experimental results.**
>
> To understand why IBL rules outperform semantic relevance-based rules (non-IBL rules), we investigate the quality of each rule. More concretely, we show the average precision and support [1] of each collected rule for different rule types below.
>
> ||FB15k237||WN18RR||UMLS||Kinship||
> |-|:-:|:-:|:-:|:-:|:-:|:-:|:-:|-|
> ||support|prec.|support|prec.|support|prec.|support|prec.|
> |IBL Rule|**708.26**|**3.74%**|**2374.28**|**12.7%**|**3.04**|**11.64%**|**8.65**|**11.58%**|
> |Non-IBL Rule|281.36|1.70%|188.29|4.92%|2.99|9.52%|6.71|5.09%|
>
> The results show that both the average precision and support of IBL rules are substantially higher than those of non-IBL rules. The support in WN18RR is even about one order of magnitude higher. The findings explain why IBL rules outperform non-IBL rules despite occupying only a small fraction of the entire rule space (Table 7, 8). Therefore, while interpreting rule-based reasoning, the instance-based equivalence relationship by IBL rules should not be overlooked.
>
> [1] Galárraga, Luis, et al. "Fast rule mining in ontological knowledge bases with AMIE++." The VLDB Journal 24.6 (2015).
>
> > **Q3: The assumption is a bit in line with collaborative filtering in the recommendations. Have you thought of using matrix factorization?**
>
> We have also considered using matrix factorization. But we found that it has less theoretical soundness than translational models. We elaborate this below:
>
> Consider the canonical decomposition of the KB: $h \otimes r \otimes t = 1$ or $0$ indicates whether $(h,r,t)$ is a valid triple. For $(h,r,?)$, $p$ is the prototype means
>
> $h \otimes r \otimes t = 1$ AND $p \otimes r \otimes t = 1$.
>
> Since $t$ has infinite solutions, $p$ cannot be explicitly expressed under the decomposition. This is in contrast to translational models which have closed-form expressions for p in Eq. (5). The lack of expressive ability prevents us from combining the IBL and matrix factorization models as in Sec 5.
>
> >**Q4: Can the prototype-based methods generalize to unseen relations or OOD scenarios?**
>
> Indeed, as the KB rapidly grows in real applications, the ability of the model to predict previously unseen/new knowledge is crucial. Instance-based learning, as we introduced in lines 39-41, ensures high-quality reasoning as the KB grows dynamically. For example, when predicting Jill Biden's lived city, even if Jill Biden moves to another city (possibly a new entity) in the future, we can still use Joe Biden as the prototype to make predictions. This new entity cannot be predicted by traditional knowledge graph embedding models. This verifies the effect of instance-based learning for new knowledge.

---

> > ### Author Response · Authors · 2022-08-07
> > **A further intuitive explanation of Q2**
> >
> > We would like to further explain the **intuition** about why IBL rules are of higher quality than non-IBL rules, especially in terms of support. This is due to the fact that the real KBs satisfy the **low-rank assumption** in the instance-based setting (line 39-43).
> >
> > More specifically, as stated in line 191-194, the IBL rule actually describes that "if $x$ and $x'$ have the same $rel_1$, then they tend to have the same $rel_0$". An IBL rule holds whenever $x$ and $x'$ have two relations ($rel_0$, $rel_1$) with the same values.  Real KBs often have a large number of entities with identical relations and values. Continuing with the example in Fig. 1, the first rule means that two entities that win the same awards also tend to have the same professions. In FB15k-237, we found 100 entities with the profession of `actor` and won the `Screen Actors Guild Award`. As a result, it is easy for the IBL rule to hold in real KBs due to the low-rank assumption.

---

> > ### Author Response · Authors · 2022-08-09
> > **A further theoretical explanation of Q2.**
> >
> > Encouraged by Reviewer DVJo, we employ translational models to provide a theoretical proof for the effectiveness of IBL rules. We will show that IBL rules always hold true under the assumption of translational models. We take TransE as an example. Consider an optimal TransE model such that $\lVert e_h + r - e_t \rVert = 0$ iff $ (h,r,t) \in KB$.
> >
> > *(The effectiveness of IBL rules)* $\forall r_0, r_1$, the IBL rule $r_0 \land r_1 \land r_1^{-1} \Rightarrow r_0$ always holds.
> >
> > **Proof:** For all entities $a$, $b$, $c$, and $d$ that satisfy the premise of the IBL rule (i.e., $(a, r_0, b), (b, r_1, c), (c, r_1^{-1}, d) \in KB$) , we have:
> >
> > $\lVert e_a + r_0 - e_b \rVert = 0, \lVert e_b + r_1 - e_c \rVert = 0, \lVert e_c + r_1^{-1} - e_d \rVert = 0 $,
> >
> > Therefore, $e_d = e_a + r_0，\lVert e_a + r_0 - e_d \rVert = 0$
> >
> > As a result,  $(a, r_0, d) \in KB$, which indicates that the hypothesis of the IBL rule also holds. So the IBL rule $r_0 \land r_1 \land r_1^{-1} \Rightarrow r_0$ always holds.
> >
> > Obviously, the above proof can be generalized to other translational models which represent inversion and compositionality (e.g., TransR, RotatE). On the other hand, semantic relevance-based rules (non-IBL rules) do not always hold (e.g., $r_0 \Rightarrow r_1$). The above proof demonstrates the effect of the IBL rule and explains why IBL rules have higher empirical support/precision.

---

> ### Author Response · Authors · 2022-08-09
> **We are looking forward to continuing to communicate with you.**
>
> Dear Reviewer iZrV,
>
> We sincerely thank you for acknowledging that the proposed idea is well-motivated, the closed-form solution is interesting, and our work connects instance-based learning and rule-based learning and encourages the rethinking of interpretability of rule-based reasoning.
>
> We have also carefully considered your questions and suggestions and given our responses. **We are looking forward to continuing to communicate with you. If you have any further questions, please let us know.** Thank you.

---

### Meta-Review · Area_Chair_nahF · 2022-08-29

**Recommendation:** Accept
**Confidence:** Certain

**Metareview:**

Develops a new paradigm for knowledge base completion, based around instance based learning.   The paper has motivation, some supporting theory and good empirical work.

Reviewers ZosV and DVJo mention a relationship with GNN/GCNs which should be further discussed in the paper.
The paper has some simple grammar and spelling errors that should be fixed up, though no reviewers mentioned this.
The authors have some enlightening discussion with reviewers, for instance on definitions, that should be included in the paper.

**Award:**

No

---

### Decision · Program_Chairs · 2022-09-14

Accept